# Lipschitz Bandits with Stochastic Delayed Feedback

**Zhongxuan Liu**
Department of Statistics
University of California, Davis
Davis, CA 95616
bkbliu@ucdavis.edu

**Yue Kang**
Microsoft
Redmond, WA 98052
yuekang@microsoft.com

**Thomas C. M. Lee**
Department of Statistics
University of California, Davis
Davis, CA 95616
tcmlee@ucdavis.edu

## Abstract

The Lipschitz bandit problem extends stochastic bandits to a continuous action set defined over a metric space, where the expected reward function satisfies a Lipschitz condition. In this work, we introduce a new problem of Lipschitz bandit in the presence of stochastic delayed feedback, where the rewards are not observed immediately but after a random delay. We consider both bounded and unbounded stochastic delays, and design algorithms that attain sublinear regret guarantees in each setting. For bounded delays, we propose a delay-aware zooming algorithm that retains the optimal performance of the delay-free setting up to an additional term that scales with the maximal delay $\tau_{\max}$. For unbounded delays, we propose a novel phased learning strategy that accumulates reliable feedback over carefully scheduled intervals, and establish a regret lower bound showing that our method is nearly optimal up to logarithmic factors. Finally, we present experimental results to demonstrate the efficiency of our algorithms under various delay scenarios.

## 1 Introduction

Multi-armed Bandit (MAB) (Auer et al., 2002) is a foundational framework for sequential decision-making under uncertainty, with widespread applications in clinical studies (Villar et al., 2015), online recommendation (Li et al., 2010), and hyperparameter tuning (Ding et al., 2022). In the standard multi-armed bandit formulation, the learner repeatedly selects an arm and immediately observes a stochastic reward drawn from an unknown distribution, using this feedback to update their policy in real time to maximize the overall cumulative reward. However, in many real-world applications, the reward signal is not always available immediately. For example, in online advertising, user engagement data such as clicks or conversions may come back to the engine minutes or even hours after an item is shown; in clinical trials, the effect of a treatment on a patient's health may be delayed and take weeks to observe. This motivates the study of bandit problems with delayed feedback, where the learner must make decisions without knowing when past rewards will arrive, introducing additional uncertainty not only in the stochastic reward values but also in the timing of their observations. To deal with this real-world challenge, recent progress has extended classical bandit algorithms to accommodate stochastic delays under multiple bandit frameworks, including MABs (Joulani et al., 2013; Vernade et al., 2017), linear bandits (Vernade et al., 2020) and kernelized bandits (Vakili et al., 2023). These approaches often rely on modifying confidence intervals or restructuring update schedules to compensate for missing feedback, and they yield robust performance under various stochastic delay models.

While delayed feedback has been extensively studied in the context of stochastic bandit problems, most existing literature focuses on simpler settings with discrete action spaces, such as the traditional MABs and contextual linear bandits. In contrast, there is little understanding of how to mitigate the

effects of delayed observations in the more challenging Lipschitz bandit framework (Kleinberg et al., 2008), where the action space is a continuous metric space and the expected reward function satisfies a Lipschitz condition. Lipschitz bandits provide a more flexible and expressive framework for a wide range of real-world applications, including hyperparameter optimization (Kang et al., 2024), dynamic pricing, and auction design (Slivkins et al., 2019). State-of-the-art algorithms for Lipschitz bandits, such as the Zooming algorithm (Kleinberg et al., 2019), leverage adaptive partitioning and activation schemes to concentrate exploration on more promising regions, achieving optimal regret bounds in the delay-free setting. However, these methods fundamentally rely on prompt reward feedback to refine partitions and update confidence estimates. In the presence of stochastic delays, they may struggle to accurately track arm performance and quality, leading to degraded performance both theoretically and empirically. Moreover, the continuous nature of the arm space amplifies the impact of missing or stale feedback, as each sampled point represents a neighborhood region whose estimation depends on delayed observations. To the best of our knowledge, the Lipschitz bandit problem with stochastic delayed feedback has not been previously explored, and poses distinct challenges that require novel algorithmic designs due to the dual complexity of delayed observations and continuous action spaces.

We take the first step toward understanding Lipschitz bandit problem in the presence of stochastic delayed feedback, and our contributions can be summarized as follows:

- For bounded delays, we extend the classic zooming algorithm in Kleinberg et al. (2008) to a delay-aware setting and attains $\tilde{O}\left(T^{\frac{d_z+1}{d_z+2}} + \tau_{\max}T^{\frac{d_z}{d_z+2}}\right)$ regret bound ($d_z$ is the zooming dimension), recovering the optimal regret bound of Lipschitz bandit (Kleinberg et al., 2019), with an additional term scales with the maximal delay $\tau_{\max}$.

- For unbounded delays, we propose a novel phased learning algorithm named Delayed Lipschitz Phased Pruning (DLPP) that accumulates reliable feedback over carefully designed intervals, and we show that the algorithm enjoys the same regret rate compared to their delay-free version, with an additive dependence on the quantiles of the delay distribution. To complement our upper bounds, we also establish the instance-dependent lower bounds in the general unbounded-delay setting, showing that our regret bound is nearly optimal up to logarithmic factors.

- We validate our theoretical findings with empirical results, showing that our methods retain the sublinear regret rate and are highly efficient under various reward functions and delay settings.

## 2 RELATED WORK

**Lipschitz Bandits**  The Lipschitz bandit problem, also known as continuum-armed bandits, was first introduced in Agrawal (1995). The problem was then extended to the general metric space in Kleinberg et al. (2008). A common approach to solving Lipschitz bandit problems is to discretize the continuous arm spac, either uniformly or adaptively, and then apply standard finite-armed bandit algorithms to deal with the exploration-exploitation tradeoff. This reduction enables the use of strategies such as the Upper Confidence Bound (UCB) method (Kleinberg et al., 2019), Thompson Sampling (TS) (Vernade et al., 2020) and elimination-based techniques (Feng et al., 2022). Beyond the standard stochastic setting, Kang et al. (2024); Nguyen et al. (2025) studied the non-stationary Lipschitz bandit problem, and Podimata & Slivkins (2021) adapted the discretization-based framework to handle adversarial rewards case. Moreover, Kang et al. (2023) investigated the Lipschitz bandits with adversarial corruptions to bridge the gap between purely stochastic and fully adversarial regimes. In addition, several extensions of the Lipschitz bandit framework have also been explored. For instance, bandits with heavy-tailed rewards are studied in Lu et al. (2019), while contextual Lipschitz bandits have been addressed in Slivkins et al. (2019); Krishnamurthy et al. (2020), and quantum variants of Lipschitz bandits have been recently investigated in Yi et al. (2025).

**Bandits with Delayed Feedback**  The stochastic multi-armed bandit (MAB) problem with randomized delays has been extensively studied in the literature. Joulani et al. (2013) analyzed how delays affect regret in online learning through partial monitoring settings and present a direct modification of the UCB1 algorithm, showing that delays lead to multiplicative regret increases in adversarial settings and additive increases in stochastic ones. Vernade et al. (2017) investigated delayed conversions in stochastic MABs, allowing for censored feedback under the assumption of known delay distributions. Variants involving delayed, aggregated, and anonymous feedback were addressed in Pike-Burke et al.

(2018), where a phase-based elimination algorithm based on the Improved UCB algorithm by Auer & Ortner (2010) was proposed. Unrestricted and potentially unbounded delay distributions have been explored in both reward-independent and reward-dependent settings in Lancewicki et al. (2021) by adapting quantile function of the delay distribution into the regret bound. Recent work have extended delayed feedback setting to a variety of bandit problems, such as linear bandits (Vernade et al., 2020), contextual MAB (Arya & Yang, 2020), generalized (contextual) linear bandits (Zhou et al., 2019; Howson et al., 2023), adaptivity and confounding in MAB (Qin & Russo, 2022), delayed payoff in MAB (Schlisselberg et al., 2025), Thompson Sampling (Wu & Wager, 2022; McDonald et al., 2023), kernel bandits (Vakili et al., 2023), Bayesian optimization (Verma et al., 2022), dueling bandits (Yi et al., 2024), adversarial bandits (Zimmert & Seldin, 2020) and best-of-both-worlds algorithms (Masoudian et al., 2024). However, the study of Lipschitz bandits under delayed feedback remains an unaddressed open problem due to the unique challenges posed by the continuous arm space, where existing methods for handling delay fail.

## 3 PROBLEM SETTINGS AND PRELIMINARIES

We study the problem of Lipschitz bandit with delayed feedback. Formally, a Lipschitz bandit problem is defined on a triplet $(\mathcal{A}, \mathcal{D}, \mu)$, where the action space $\mathcal{A}$ is a compact doubling metric space equipped with metric $\mathcal{D}$, both known to the agent. The unknown expected reward function $\mu : \mathcal{A} \to [0, 1]$ is a 1-Lipschitz function defined on $\mathcal{A}$ w.r.t. the metric $\mathcal{D}$, i.e. $|\mu(x_1) - \mu(x_2)| \leq \mathcal{D}(x_1, x_2), \quad \forall x_1, x_2 \in \mathcal{A}$. We refer to the triple $(\mathcal{A}, \mathcal{D}, \mu)$ as an instance of the Lipschitz bandit problem. And without loss of generality, we assume that the diameter of $(\mathcal{A}, \mathcal{D})$ is bounded by 1, which is a standard assumption as in Kleinberg et al. (2019).

At each time round $t \in [T] := \{1, 2, \ldots, T\}$, the agent pulls an arm $x_t \in \mathcal{A}$ and a stochastic reward sample $y_t = \mu(x_t) + \epsilon_t$ is generated, where $\epsilon_t$ is a sub-Gaussian i.i.d. white noise with sub-Gaussian parameter $\sigma$ conditional on filtration $\mathcal{H}_t^0 = \{(x_s, y_s) \mid s \in [T-1]\}$. Without loss of generality, we assume that $\sigma = 1$ throughout the remainder of the analysis for simplicity. In the classical setting, the agent observes the full feedback immediately after each round, so the available information at the beginning of round $t$ is captured by the filtration $\mathcal{H}_t^0$, which includes all past actions and rewards up to round $t-1$. In contrast, under stochastic delayed feedback, the agent does not observe the reward $y_t$ at the end of round $t$, but only after a random delay $\tau_t$ where $\tau_t$ is a nonnegative integer drawn from an unknown delay distribution $f_\tau$. The agent receives delayed feedback in the form of the pair $(x_t, y_t)$, without access to the original round index $t$ or the delay value $\tau_t$, making it impossible to directly associate the feedback with the round in which the action was taken. The delay is supported in $\mathbb{N} \cup \{\infty\}$, where the infinite delay corresponds to the case where the reward is never observed (i.e., missing feedback). We assume that delays are independent of both the chosen arm and the realized reward. Now we define the observed filtration $\mathcal{H}_t$ as

$$\mathcal{H}_t = \{(x_s, y_s) \mid s + \tau_s \leq t - 1\} \cup \{(x_s, y_s) \mid s \leq t - 1, s + \tau_s \geq t\},$$

and $\mathcal{H}_t$ is the information available at the beginning of round $t$ to the agent.

Let $Q : [0, 1] \to \mathbb{N}$ denote the quantile function of the delay distribution, that is,

$$Q(p) = \min\{n \in \mathbb{N} \mid P(\tau \leq n) \geq p\}.$$

where $\tau$ is the random delay. Note that we only take integer value as the function values, since we assume the delay is supported in $\mathbb{N} \cup \{\infty\}$. Similar to most bandit learning problems, the goal of the agent is to minimize the cumulative regret, defined as

$$R(T) = \sum_{t=1}^{T} (\mu^* - \mu(x_t)) = \mu^* \cdot T - \sum_{t=1}^{T} \mu(x_t), \quad \text{where} \quad \mu^* = \max_{x \in \mathcal{A}} \mu(x).$$

The loss (optimality gap) of arm $x$ is defined as $\Delta(x) = \mu^* - \mu(x)$ for $x \in \mathcal{A}$. Notably, we consider the cumulative regret of all generated rewards rather than the rewards observed by time horizon $T$.

An important pair of concepts in Lipschitz bandits for a problem instance $(\mathcal{A}, \mathcal{D}, \mu)$ are the covering dimension $d$ and zooming dimension $d_z$ (Kleinberg et al., 2008). Let $B(x, r) := \{x' \in \mathcal{A} : \mathcal{D}(x, x') \leq r\}$ denote a closed ball centered at $x$ with radius $r$ in $(\mathcal{A}, \mathcal{D})$. Let $\mathcal{S}$ be a subset of $\mathcal{A}$, then a subset $\mathcal{C}$ of $\mathcal{A}$ is a $r$-covering of $\mathcal{S}$ if $\mathcal{S} \subseteq \bigcup_{x \in \mathcal{C}} B(x, r)$. The $r$-covering number $N_{\mathrm{cov}}(r)$ of

metric space $(\mathcal{A}, \mathcal{D})$ is defined as the least number of balls with radius no more than $r$ required to completely cover $(\mathcal{A}, \mathcal{D})$, with possible overlaps between the balls. The $c$-covering dimension $d$ is defined as the smallest $d$ such that for every $r > 0$ we require only $O(r^{-d})$ balls with radius no more than $r$ to cover the metric space $(\mathcal{A}, \mathcal{D})$:

$$d = \inf \left\{ n \geq 0 : \exists c > 0, \forall r > 0, N_{\mathrm{cov}}(r) \leq cr^{-n} \right\}.$$

On the other hand, the $r$-zooming number $N_{\mathrm{zom}}(r)$ and the zooming dimension $d_z$ depend not only on the metric space $(\mathcal{A}, \mathcal{D})$ but also on the underlying payoff function $\mu$. Define the $r$-optimal region as $\{x \in \mathcal{A} : \Delta(x) \leq r\}$, where $\Delta(x) = \mu^* - \mu(x)$ denotes the suboptimality gap at point $x$. The $r$-zooming number $N_z(r)$ is the minimal number of balls of radius at most $r/16$ required to cover the $r$-optimal region. The $c$-zooming dimension $d_z$ is the smallest value $d$ such that, for all $r \in (0, 1]$, the $r$-optimal region can be covered by $O(r^{-d})$ balls of radius at most $r/16$:

$$d_z = \inf \left\{ n \geq 0 : \exists c > 0, \forall r \in (0, 1], N_{\mathrm{zom}}(r) \leq cr^{-n} \right\}.$$

Covering dimension characterized the benignness of a metric space, in terms of the least number of balls required to cover the entire space to get information, while the zooming dimension captures the difficulty of a specific problem instance, reflecting how easily the optimal arm can be distinguished from suboptimal ones. It is obvious that $0 \leq d_z \leq d$ since the $r$-optimal region is a subset of $\mathcal{A}$. Also, $d_z$ can be significantly smaller than $d$ in benign cases. For example, let $([0, 1]^n, \ell_2)$, $n \geq 1$, and $\mu \in C^2([0, 1]^n)$ with a unique $x^*$ and is strongly concave in a neighborhood of $x^*$. Then $d_z = n/2$, whereas $d = n$. Therefore, it can greatly reduce the regret bound when the problem instance is benign. However, $d_z$ is not revealed to the agent since it depends on the unknown payoff function $\mu$, and hence it would be difficult to design algorithms without the knowledge of $d_z$ while the regret bound depends on $d_z$.

## 4 Delay with Bounded Support: Delayed Zooming Algorithm

We first consider the case of bounded delays. Specifically, we assume there exists a positive integer $\tau_{\max}$ such that the delay $\tau_t$ is supported on $\{0, 1, \ldots, \tau_{\max}\}$. This implies that feedback is always eventually observed and never missing for all rounds prior to $t = T - \tau_{\max}$, with a maximum delay of $\tau_{\max}$ rounds. To address this setting, we propose a delay-aware variant of the zooming algorithm, called the Delayed Zooming algorithm, which maintains the optimal regret rate of the delay-free setting up to an additive term that scales with the maximal delay. The algorithm achieves a regret bound of order $\tilde{O}\left(T^{\frac{d_z+1}{d_z+2}} + \tau_{\max} T^{\frac{d_z}{d_z+2}}\right)$.

We now introduce some additional notations used in the algorithm. At the beginning of each round $t$, for any arm $x \in \mathcal{A}$, $n_t(x)$ is the number of times arm $x$ has been pulled, $v_t(x)$ is the number of times that rewards generated by arm $x$ has been observed, and $w_t(x)$ is the number of missing observations of arm $x$ up to time $t$. By definition, we have $n_t(x) = v_t(x) + w_t(x)$. Let $\mu_t(x)$ be the sample average reward using the observed samples at time round $t$, and $r_t(x)$ be the confidence radius such that with high probability $\mu_t(x)$ concentrates around its expectation $\mu(x)$ with radius $r_t(x)$, i.e. $|\mu(x) - \mu_t(x)| \leq r_t(x)$, and $B_t(x) = B(x, r_t(x))$ be the confidence ball of arm $x$.

Similar to the classic zooming algorithm, our delayed variant leverages the UCB principle and adaptive discretization (Slivkins et al., 2019). It maintains an active set of arms such that all arms in $\mathcal{A}$ are covered by the confidence balls of active arms. If an arm is not covered, it is activated. This mechanism allows the algorithm to adaptively focus on high-reward regions. In each round, the agent selects the active arm with the highest index $I_t(x) = \mu_t(x) + 2r_t(x)$ (breaking ties arbitrarily) and pulls it. The confidence radius $r_t(x)$ is defined below based on the confidence level $\delta$.

$$r_t(x) = \sqrt{\frac{4 \log T + 2 \log(2/\delta)}{1 + v_t(x)}}.$$

In the classic zooming algorithm, the agent updates the empirical mean reward and the corresponding confidence radius immediately after pulling an arm. However, in the presence of delay, these updates cannot be performed in real time, as the agent lacks access to both unobserved rewards and the number of pending pulls for each arm. Inspired by the idea of Delayed-UCB1 (Joulani et al., 2013), we modify the classic zooming algorithm by replacing $n_t(x)$, defined as the number of times arm $x$

has been pulled, with $v_t(x)$, the number of observed rewards from arm $x$, and compute the empirical mean using only the available feedback. While this modification appears direct and natural given the prior literature on bandits with delayed feedback (Vernade et al., 2017), we highlight that the analysis is far from trivial and significantly different from that of multi-armed bandit with delays and the classic zooming algorithm. Specifically, the regret analysis of the classic zooming algorithm relies on a key result that bounds the sub-optimality gap by the confidence radius, i.e. $\Delta(x) \leq 3r_t(x)$, which ensures that suboptimal arms are not pulled too frequently. This bound holds in the delay-free setting because $r_t(x)$ only changes when arm $x$ is played. Under delayed feedback, however, $r_t(x)$ can shrink over time even when the arm is not being pulled, due to the arrival of delayed observations. This can cause the confidence radius to decrease too rapidly. In contrast, the extension from `UCB1` to `Delayed-UCB1` does not have this complication since its analysis on MAB is simpler and does not depend on such a suboptimality gap inequality. To address this challenge, we develop a "lazy update" mechanism showing that, under bounded delays, a similar guarantee still holds: the sub-optimality gap is bounded by $6r_t(x)$. Establishing this result is not a straightforward extension of the original proof and requires careful treatment of the delayed information flow, yet it notably preserves the theoretical guarantee of the classic zooming algorithm up to a constant factor.

**Lazy Update Mechanism:** We introduce a "lazy update" mechanism here to address the issue of decreasing confidence radius caused by delayed feedback when an active arm has not been pulled. For each active arm $x$, we maintain a cache queue $Q[x]$ to store excessive delayed feedback from arm $x$, and record $v_s(x)$, where $s < t$ is the last time the agent pulls arm $x$ and $t$ is the current round. The cache $Q[x]$ is cleared immediately when the agent pulls arm $x$ again, thereby reactivating the suboptimality gap bound for arm $x$. The caches for other active arms remain unchanged. Specifically, suppose at time $s$ the agent pulls arm $x$, then at any time $t > s$ while updating $v(x)$ and $\mu(x)$, if $v_t(x) + 1 \leq 4v_s(x)$, the algorithm fetch scheduled feedback and update $\mu(x)$ and $r(x)$; if $v_t(x) + 1 > 4v_s(x)$, the algorithm cache any incoming feedback of $x$ and do not update for arm $x$, until the arm $x$ is pulled again. This lazy update rule ensures that the confidence radius associated with arm $x$ does not fall below half of its value from the last time the arm was pulled. This mechanism is critical for establishing the suboptimality gap bound lemma 8.

The complete description is given in Algorithm 1. As in the classic zooming algorithm, our delayed zooming algorithm also requires a covering oracle (i.e. line 5 of Algorithm 1) which takes a finite collection of closed balls and either declares that they cover all arms in the metric space or outputs an uncovered arm to apply the activation rule (Kleinberg et al., 2019). Note that if the delays are zero, this algorithm reduces to the original zooming algorithm.

In the following Theorem 1, we show that the Algorithm 1 enjoys the same regret rate compared to its non-delayed version, with an additive penalty depending on the maximum delay.

**Theorem 1** (Regret Bound for Delayed Zooming algorithm). *Consider an instance $(\mathcal{A}, \mathcal{D}, \mu)$ of the delayed Lipschitz Bandits problem with time horizon $T$ and a delay distribution $f_\tau$ with bounded support such that $P(\tau \leq \tau_{max}) = 1$. For any given problem instance, with probability at least $1 - \delta$, the delayed zooming algorithm attains regret*

$$R(T) \leq O\left(T^{\frac{d_z+1}{d_z+2}}\left(c\log\frac{T}{\delta}\right)^{\frac{1}{d_z+2}} + c\tau_{\max} \cdot \left(\frac{T}{\log T}\right)^{\frac{d_z}{d_z+2}}\right) = \tilde{O}\left(T^{\frac{d_z+1}{d_z+2}} + \tau_{\max}T^{\frac{d_z}{d_z+2}}\right),$$
(1)

*where $d_z$ is the c-zooming dimension of $(\mathcal{A}, \mathcal{D}, \mu)$.*

*Remark* 2. When there is no delay, i.e. $\tau_{max} = 0$, the regret bound reduces to $\tilde{O}\left(T^{\frac{d_z+1}{d_z+2}}\right)$, which is exactly the regret bound of Lipschitz bandits. When the action space is finite, i.e. $d_z = 0$, the regret bound reduces to $O(\sqrt{cT\log T} + c\tau_{\max})$, with $c$ being the number of arms, which is the regret bound of $c$-armed bandits with bounded delays in Joulani et al. (2013).

The detailed proof is provided in Appendix A. For technical reasons, our analysis assumes bounded delays, though the algorithm also performs well under unbounded delays in our experiments (Section 7). However, in practice, the delays can be unbounded or even result in missing feedback. To handle the more general settings of unbounded delay, we propose a novel phased learning strategy in the next Section 5.

---

**Algorithm 1** Delayed Zooming Algorithm

---

**Input:** Arm metric space $(\mathcal{A}, \mathcal{D})$; time horizon $T$; probability rate $\delta$.
**Initialization:** Active set $\mathscr{A} = \emptyset$.

1:  **for** $t = 1, \ldots, T$ **do**
2:      **for** any incoming scheduled feedback $(x_t, y_t)$ **do**
3:          **if** $v_t(x) + 1 \leq 4v_s(x)$ **then**
4:              Update $\mu_t(x_t)$, $r_t(x_t)$ and $v_t(x_t)$.
5:          **else**                                                    #Lazy Update
6:              push $(x_t, y_t)$ into cache $Q[x]$.
7:          **end if**
8:      **end for**
9:      **if** some arm is not covered **then**                        #Activation
10:         pick any such arm $x_t$ and add it to $\mathscr{A}$.
11:     **else**                                                       #Selection
12:         select any arm from $\mathscr{A}$ such that $x_t \in \arg\max_{x \in \mathscr{A}} I_t(x)$.
13:     **end if**
14:     Pull arm $x_t$, and schedule its reward $y_t$ at round $t + \tau_t$. Record $v_s(x_t) = v_t(x_t)$.
15:     **if** Cache $Q[x_t]$ is not empty **then**
16:         Use all feedback in $Q[x_t]$ to update $\mu_t(x_t)$, $r_t(x_t)$ and $v_t(x_t)$. Clear $Q[x_t]$.
17:     **end if**
18:     Set $t = t + 1$.
19: **end for**

---

## 5 DELAY WITH UNBOUNDED SUPPORT: DELAYED LIPSCHITZ PHASED PRUNING

In practice, the bounded delay assumption may not always hold, as feedback can be missing or censored, corresponding to an infinite delay. To address this challenge, and inspired by the success of elimination-based methods in Lipschitz bandits (Feng et al., 2022; Kang et al., 2023), we propose a novel phased learning algorithm called Delayed Lipschitz Phased Pruning (DLPP). DLPP accumulates reliable feedback over carefully scheduled intervals and achieves a regret bound that is nearly optimal up to logarithmic factors, as supported by our lower bound analysis. While BLiN (Feng et al., 2022) is the first elimination-based algorithm for Lipschitz bandits, our approach is fundamentally different in both methodology and analysis. The differences in DLPP from BLiN are summarized as below:

- We do not assume a batched communication constraint; we consider individual reward delays.
- Our method applies to any compact doubling metric space $(\mathcal{A}, \mathcal{D})$ whereas their approach is limited to $([0,1]^d, \|\cdot\|_\infty)$ and partitions the space using axis-aligned cubes.
- DLPP samples uniformly from all remaining balls, while their method repeatedly samples the same cube a fixed number of times before moving to the next.

Compared to Algorithm 1, our algorithm requires a different covering oracle which takes a subset of $\mathcal{A}$ and returns a $r$-covering of the input set to further zoom in on regions with higher empirical average rewards. The full description of the algorithm is found in Algorithm 2. We initialize with $\mathcal{C}_1$ as a $r_1$-covering of the action space $\mathcal{A}$, and let $\mathcal{B}_1 = \{B(x, r_1) \mid x \in \mathcal{C}_1\}$ be the collection of closed balls centered in the $r_1$-covering, and $r_m = 2^{-m}$ the radius sequence. Note that $\mathcal{B}_1$ covers the action space $\mathcal{A}$, and the two core steps are introduced as follows:

**Uniform Round-Robin Sampling:** In each phase $m$, our algorithm proceeds as follows. Let $\mathcal{B}_m$ be the set of currect active balls, and set the active set $\mathcal{B}_m^+ = \mathcal{B}_m$. For each ball $B \in \mathcal{B}_m^+$, we sample an arm $x \in B$ uniformly and play it, schedule the reward to be observed at time $t + \tau_t$. Then we observe any incoming scheduled reward $(C, y_C)$ and update the corresponding empirical average reward $\hat{\mu}(C)$ and number of observed times $v(C)$ of $C$. If $v(C) \geq v_m = (2\log T + \log(2/\delta))/2r_m^2$, we remove $C$ from the active set $\mathcal{B}_m^+$, and do not play it anymore in this phase (i.e. line 6 of Algorithm 2). Increment the timer $t = t + 1$ and exit the algorithm if $t > T$. The number $v_m$ is carefully designed such that the empirical average reward over a ball concentrated over its true mean with high probability. We assume that tuple $(C, y_C)$ is observed as feedback. It is necessary to know which ball an arm

---

**Algorithm 2** Delayed Lipschitz Phased Pruning (DLPP)

---

**Input:** Arm metric space $(\mathcal{A}, \mathcal{D})$; time horizon $T$; probability rate $\delta$.
**Initialization:** Timer $t = 1$; Phase counter $m = 1$; Radius sequence $r_m = 2^{-m}$; $\mathcal{C}_1$: a $1/2$-covering
   of $\mathcal{A}$; $\mathcal{B}_1 := \{B(x, r_m) \mid x \in \mathcal{C}_1\}$.
1: **while** $t \leq T$ **do**
2:     Set $\mathcal{B}_m^+ = \mathcal{B}_m$. For each ball $B \in \mathcal{B}_m^+$, set $v(B) = 0, \hat{\mu}_m(B) = 0$.
3:     **while** $\mathcal{B}_m^+ \neq \emptyset$ and $t \leq T$ **do**
4:       **for** $B \in \mathcal{B}_m^+$ **do**                   `#Uniform Round-Robin Sampling`
5:         Sample an arm $x \in B$ and play it. Schedule the reward to be observed at time $t + \tau_t$.
6:         **for** any incoming scheduled reward $(C, y_C)$ **do**
7:           Update $\hat{\mu}_m(C) = (\hat{\mu}_m(C) \cdot v(C) + y_C)/(v(C) + 1)$ and $v(C) = v(C) + 1$.
8:           If $v(C) \geq v_m = (2 \log T + \log(2/\delta))/2r_m^2$, remove $C$ from $\mathcal{B}_m^+$.
9:         **end for**
10:         Set $t = t + 1$. If $t > T$, break.
11:       **end for**
12:     **end while**
13:     Let $\hat{\mu}_m^* = \max_{B \in \mathcal{B}_m} \hat{\mu}_m(B)$. For each ball $B \in \mathcal{B}_m$, remove $B$ from $\mathcal{B}_m$ if $\hat{\mu}_m^* - \hat{\mu}_m(B) \geq$
   $8r_m$. Let $\mathcal{B}_m^*$ be the set of balls not eliminated.                `#Elimination`
14:     For each ball in $B \in \mathcal{B}_m^*$, find a $r_m/2$-covering $\mathcal{C}$, and set $\mathcal{C}_{m+1} = \mathcal{C} \bigcup \mathcal{C}_{m+1}$. Set $\mathcal{B}_{m+1} =$
   $\{B(x, r_{m+1}) \mid x \in \mathcal{C}_{m+1}\}$.                         `#Discretization`
15:     Set $m = m + 1$.
16: **end while**

---

$x$ belongs to, as the balls may overlap; however, identifying the specific arm within a ball is not required, since all arms in a ball are treated uniformly to represent the ball and sampled at random. The procedure proceeds in a Round-Robin fashion such that we eliminate the balls with low empirical average reward in the next step.

**Active Region Pruning and Discretization:** When we have sufficient feedback for each active ball, the algorithm performs pruning on the current active region. Let $\hat{\mu}_m^* = \max_{B \in \mathcal{B}_m} \hat{\mu}_m(B)$ denote the highest current empirical average reward. We eliminate any ball $B \in \mathcal{B}_m$ whose empirical mean deviates significantly from $\hat{\mu}_m^*$ according to the following pruning rule: a ball $B \in \mathcal{A}_m$ is eliminated (removed from $\mathcal{B}_m$) if $\hat{\mu}_m^* - \hat{\mu}_m(B) \geq 8r_m$ (i.e. line 9 of Algorithm 2). As the algorithm progresses, the radius $r_m$ shrinks exponentially, making the pruning criterion increasingly stringent in later phases. This ensures that, with high probability, only balls containing near-optimal arms survive. Let $\mathcal{B}_m^*$ denote the set of surviving balls after pruning. We denote the set of surviving balls as $\mathcal{B}_m^*$. Following pruning, we refine the surviving regions by further discretizing each promising ball in $\mathcal{B}_m^*$. Specifically, for each $B \in \mathcal{B}_m^*$, we compute a $r_m/2$-covering and add the centers to the new covering set $\mathcal{C}_{m+1}$. We then construct the next collection of active balls as $\mathcal{B}_{m+1} = \{B(x, r_{m+1}) \mid x \in \mathcal{C}_{m+1}\}$ (see line 10 of Algorithm 2), and proceed to the next phase $m + 1$.

In the following Theorem 3, we show that the Algorithm 2 enjoys the same regret rate compared to their non-delayed version, with an additive dependence on the quantiles of the delay distribution.

**Theorem 3** (Regret Bound of Delayed Lipschitz Phased Pruning). *Consider an instance $(\mathcal{A}, \mathcal{D}, \mu)$ of the delayed Lipschitz Bandit problems with time horizon $T$ and a delay distribution with quantile function $Q(p)$, with probability at least $1 - 2\delta$, the Algorithm 2 attains regret*

$$R(T) \lesssim \min_{p \in (0,1]} \left\{ \frac{1}{p} T^{\frac{d_z+1}{d_z+2}} \left( c \log \frac{T}{\delta} \right)^{\frac{1}{d_z+2}} + Q(p). \right\}, \tag{2}$$

*where $d_z$ is the zooming dimension of $(\mathcal{A}, \mathcal{D}, \mu)$.*

The proof of Theorem 3 is highly non-trivial and consists of several core steps. The detailed proof is provided in Appendix B.

*Remark* 4. The claimed bound is achieved minimizing over a single quantile $p \in (0, 1]$, in the sense that it reaches the smallest order. When there is no delay, i.e. $\tau_{max} = 0$, the regret bound reduces to $\tilde{O}\left(T^{\frac{d_z+1}{d_z+2}}\right)$, which is exactly the regret bound of Lipschitz bandits; this is the same as in Theorem 1.

The increase in regret due to delays in the claimed bound does not scale with the underlying zooming dimension $d_z$. Take $p = 0.5$, since $\tau_{med} = Q(0.5)$ denotes the median of the delays, the bound becomes $\tilde{O}\left(T^{\frac{d_z+1}{d_z+2}} + \tau_{med}\right)$, which is essentially the same with the order of Theorem 1. This quantile-based bound provides a flexible characterization of the delay distribution's effect on regret, allowing the algorithm to adapt to the effective central mas of delays rather than worst-case outliers. To see why the increase in regret due to delays should scale with a certain quantile, one can simulate a black box algorithm for the rounds that delay is smaller than $\tau_{med}$, and take the last action of the black box algorithm for the rest rounds (approximately half of the rounds). Since the rewards are stochastic and independent of both time and delay, the regret incurred on rounds with delays greater than $\tau_{med}$ is comparable to the regret of the black-box algorithm on the remaining rounds, as a result, the total regret is essentially twice that of the black-box algorithm (Lancewicki et al., 2021).

## 6    LOWER BOUND

Next, we present an instance-dependent lower bound for Lipschitz bandits with stochastic delays.

**Theorem 5.** *Consider the delayed Lipschitz Bandit problem with a delay distribution with quantile function $Q(\cdot)$. Fix an arbitrary time horizon $T$, there exists at least one instance such that the regret on that instance satisfies:*

$$R(T) \gtrsim \frac{T^{\frac{d_z+1}{d_z+2}}(c\log T)^{\frac{1}{d_z+2}}}{p\log T} - \frac{1}{p} + \bar{\Delta} \cdot Q(p) \tag{3}$$

*for sufficiently large T, where $\bar{\Delta} = \int_{\mathcal{A}} \Delta(x) / \int_{\mathcal{A}} 1$.* [1]

The lower bound is proved using a specified delay distribution such that the delay is a fixed value $\tau_0$ with probability $p$, and $\infty$ otherwise (missing feedback). The first term in Eq. (3) retains the optimal regret rate of Lipschitz bandits, up to logarithmic factors, scaling with the delay quantiles. The first term also matches the existing lower bound, proved through a reduction from the non-delayed lower bound for Lipschitz bandits (Kleinberg et al., 2019; Slivkins, 2011). The core idea is to simulate a delayed variant of a Lipschitz bandit algorithm by introducing a Bernoulli sampling mechanism with parameter $p$. At each round, a Bernoulli random variable is sampled; if its outcome is 1, the original Lipschitz bandit algorithm selects the action, thereby mirroring the behavior of its delayed counterpart. Otherwise, only the delayed version takes an action. This probabilistic coupling ensures that the original and delayed algorithms align with probability $p$, enabling a controlled simulation of delay effects. Compared to the upper bound in Theorem 3, there is an additional term. Specifically, the last term in Eq. (3) arises from the fact that the algorithm receives no feedback during the first $\tau_0 = Q(p)$ rounds. As a result, the learner is unable to distinguish between arms in this initial period, and there exists at least one problem instance in which the learner incurs a regret of $\tau_0 \bar{\Delta}$ where $\bar{\Delta} = \int_{\mathcal{A}} \Delta(x) / \int_{\mathcal{A}} 1$ per round over the first $\tau_0$ rounds. This demonstrates that the lower bound essentially matches the upper bound, implying that our regret bound is nearly optimal up to logarithmic factors. A more detailed discussion and complete proof are provided in Appendix C.

## 7    EXPERIMENTS

In this section, we present simulation results to demonstrate the performance of our proposed algorithms under various delay conditions. To the best of our knowledge, there are no existing algorithms that directly address the Lipschitz bandit problem with stochastic delayed feedback. Although the `BLiN` algorithm proposed in Feng et al. (2022) shares some structural similarities with our DLPP algorithm, it is not applicable in our setting. In particular, `BLiN` relies on immediate or batched feedback, whereas in our model, the presence of additional delays and the absence of a batched communication constraint render region-level reward estimates unreliable.

To thoroughly validate the robustness of our proposed methods, we employ three types of expected reward functions under both bounded and unbounded delay conditions, each with varying average delays. We first consider the metric space $([0, 1], |\cdot|)$ with two expected reward functions that behave

---

[1] $\bar{\Delta}$ is well-defined since $\mathcal{A}$ is a compact doubling metric space and therefore it is a measure space that supports a doubling measure.

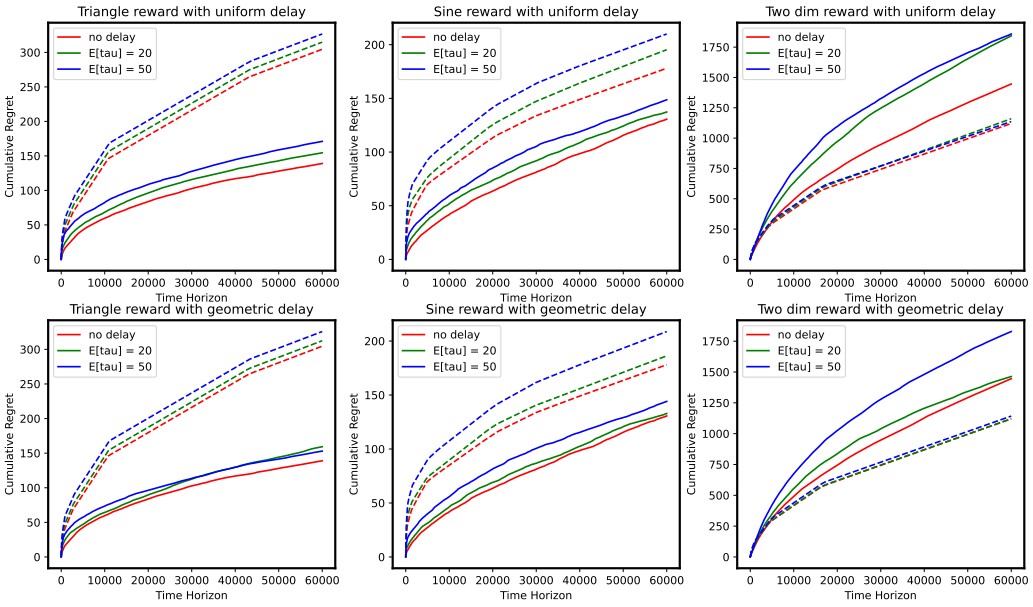

Figure 1: Plots of cumulative regrets of Delayed Zooming algorithm (solid line) and DLPP (dashed line) under different settings with three levels of average delay: no delay (red), $\mathbb{E}[\tau] = 20$ (green) and $\mathbb{E}[\tau] = 50$ (blue). The first row corresponds to uniform distribution for the bounded delays, and the second row corresponds to geometric distribution for the unbounded delays. The three columns correspond to the triangle, sine, and two-dimensional reward function (from left to right).

differently around their maxima, including a triangle function $\mu(x) = 0.8 - 0.9 \cdot |x - 0.4|$, and a sine wave function $\mu(x) = \frac{2}{3} \left| \sin\left(\frac{5\pi}{3} x\right) \right|$ with two different maximum point ($x = 0.3, 0.9$). We further consider the more complicated arm metric space ($[0,1]^2, \|\cdot\|_\infty$), and the expected reward function is a two dimensional function $\mu(x) = 1 - 0.7 \|x - x_1\|_2 - 0.4 \|x - x_2\|_2$, where $x_1 = (0.7, 0.8)$ and $x_2 = (0, 0.1)$, with $\mu^* = \mu(x_1)$. We set with time horizon $T = 60,000$ and false probability rate $\delta = 0.01$, and we repeat the experiment over independent trials $B = 30$ and take the average cumulative regret. We evaluate the algorithms with different delay distributions. In the first case, we consider the uniform distribution for the bounded delay distributions, whereas in the second case, we implement the geometric distribution for the unbounded delays. As in the empirical study done by Chapelle (2014), delays are empirically shown to have an exponential decay. We consider both cases with different mean that $\mathbb{E}[\tau] \in \{20, 50\}$. Since there is no existing delayed Lipschitz bandit algorithm, we also include the result of the delay-free setting as the baseline to compare the performance. The graphic regret curves are reported in Figure 1.

As shown in Figure 1, both of our proposed algorithms exhibit sublinear cumulative regret under both bounded and unbounded delay settings. Compared to the regret curves in the non-delayed case, the regret under delayed feedback remains on the same scale, which is consistent with the theoretical regret bounds that larger expected delays lead to higher cumulative regret. While the regret of Delayed Zooming algorithm is lower for one dim reward functions, it is higher for the two dim reward functions. Also, for the two dim reward functions, DLPP is more efficient than the Delayed Zooming due to its pruning and discretization strategy. It is worth noting that although we show that the Delayed Zooming algorithm works for bounded delays in Section 4, our empirical results show that it also works for unbounded delays, leaving a compelling open problem to extend the Delayed Zooming algorithm to unbounded delays. Additionally, unlike the Delayed Zooming algorithm, the regret curve of DLPP appears approximately piecewise linear. This behavior arises because DLPP samples arms uniformly from each surviving ball within a given phase while the surviving active balls are those with promising rewards, resulting in phase-wise regret accumulation. Due to space constraints, we defer additional experimental results to Appendix D.

## 8 CONCLUSION

In this work, we introduce the problem of Lipschitz bandits under stochastic delayed feedback and propose two algorithms that address bounded and unbounded delays, respectively, supported by comprehensive theoretical analysis. For the bounded-delay setting, we propose a delay-aware zooming algorithm called Delayed Zooming Algorithm that matches the optimal performance of the delay-free case (Kleinberg et al., 2008) up to an additive term that scales with the maximal delay. For unbounded delays, we propose Delayed Lipschitz Phased Pruning (DLPP) that novelly accumulates reliable feedback over carefully scheduled intervals and achieves near-optimal regret bounds, with an additional term that depends on quantiles of the delay distribution. We further establish a lower bound that nearly matches our deduced upper bounds up to logarithmic factors, demonstrating that the regret bound of DLPP is nearly optimal. The effectiveness of our proposed algorithms is finally validated through numerical experiments.

**Limitation:** Our delayed zooming algorithm requires bounded delays to recovers the optimal regret bound up to an additional term, and extending it to the unbounded delay case without additional assumptions on the delay distribution remains an open and challenging problem.

## ACKNOWLEDGEMENTS

We truly appreciate the constructive feedback from the anonymous reviewers and area chair. This work was partially supported by the National Science Foundation under grants DMS-2210388 and DMS-2515304 GFI.

ETHICS STATEMENT

We confirm adherence to the ICLR Code of Ethics. This research is based on simulated datasets, does not involve human participants, and presents no foreseeable ethical risks. All experimental procedures follow established standards for scientific integrity.

REPRODUCIBILITY STATEMENT

All theoretical results presented in this work are supported by complete proofs in the appendix. The full source code used to generate our experimental results will be included as supplementary materials to facilitate reproducibility.

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

# A  ANALYSIS OF DELAYED ZOOMING ALGORITHM

*Proof sketch.* The theorem is proved via the lemma that bounds the sub-optimality gap by the confidence radius, i.e. $\Delta(x) \leq 6r_t(x)$. This lemma essentially shows that the agent will not play the bad arm too much, hence bound $v_t(x)$. On the other hand, the lemma estabilish another lemma, such that the agent does not activate too many arms In other words, by utilizing the Lipschitzness, for any two active arms, their distance is lower bounded, allowing us to discretize the metric space.

## A.1  SOME USEFUL LEMMAS

**Definition 6.** For each arm $x$, define the clean event as

$$\mathcal{E}_x = \{\forall t \in [T], |\mu_t(x) - \mu(x)| \leq r_t(x)\} .$$

**Lemma 7.** *The probability of the clean event $\mathcal{E} = \bigcap_{x \in \mathcal{A}} \mathcal{E}_x$ is at least $1 - \delta$.*

*Proof.* For each time $t$, fix some arm $x$ that is active by the end of time $t$. Recall that each time the algorithm plays arm $x$, the reward is sampled i.i.d. from some unknown distribution $\mathbb{P}_x$ with mean $\mu(x)$. Define random variable $Z_{x,s}$ for $1 \leq s \leq v_t(x)$ as follows: for $s \leq v_t(x)$, $Z_{x,s}$ is the observed reward from the $s$-th time arm $x$ is played, and for $s > v(x)$ it is an independent sample from $\mathbb{P}_x$. For each $k \leq T$, by Hoeffding Inequality,

$$\Pr\left(\left|\mu(x) - \frac{1}{k}\sum_{s=1}^{k} Z_{x,s}\right| \leq \sqrt{\frac{4\log T + 2\log(2/\delta)}{1+k}}\right) \geq 1 - \delta T^{-2}.$$

For any $x \in \mathcal{A}$ and $j \leq T$, the event $\{x = x_j\}$ is independent of the random variables $Z_{x,s}$. Taking the union bound over all $k \leq T$, it follows that

$$\Pr\left(\forall t, |\mu_t(x) - \mu(x)| \leq r_t(x) \mid x = x_j\right) \geq 1 - \delta T^{-1}.$$

Integrating over all arms $x$ we obtain

$$\Pr[\mathcal{E}_x] = \Pr\left(\forall t, |\mu_t(x_j) - \mu(x_j)| \leq r_t(x_j)\right) \geq 1 - \delta T^{-1}.$$

Now, taking the union bound over all $j \leq T$ concludes the proof. $\square$

For what follows in this subsection, we assume clean event unless specified.

**Lemma 8.** *For each arm $x$ and each round t, we have $\Delta(x) \leq 6r_t(x)$ .*

*Proof.* Suppose that arm $x$ is played in this round. By the covering invariant, the best arm $x^*$ was covered by the confidence ball of some active arm $y$, i.e. $x^* \in B_t(y)$. By selection rule, it follows that

$$I_t(x) \geq I_t(y) = \mu_t(y) + r_t(y) + r_t(y) \geq \mu(y) + r_t(y) \geq \mu(x^*) = \mu^*.$$

The last inequality holds because of the Lipschitz condition. On the other hand,

$$I_t(x) = \mu_t(x) + 2r_t(x) \leq \mu(x) + r_t(x) + 2r_t(x) = \mu(x) + 3r_t(x).$$

Therefore, we have

$$\Delta(x) = \mu^* - \mu(x) \leq 3r_t(x).$$

Now suppose arm $x$ is not played in round $t$. If it has never been played before round $t$, then $r_t(x) > 1$ since the diameter is at most 1 and the lemma follows trivially. Otherwise, let $s$ be the round that arm $x$ was played last time. By the "lazy update" trick, $v_t(x) \leq 4v_s(x)$, and plug in the definition of the confidence radius we have $r_t(x) \geq 2r_s(x)$. Since at time $s$, arm $x$ was played by selection rule, it follows that

$$\Delta(x) \leq 3r_s(x) \leq 6r_t(x).$$

Hence for each arm $x$ and each round $t$, we have $\Delta(x) \leq 6r_t(x)$, as desired. $\square$

**Corollary 9.** *For any two active arms $x, y$, we have $\mathcal{D}(x, y) > \frac{1}{6}\min(\Delta(x), \Delta(y))$.*

*Proof.* Without loss of generality, assume $x$ has been activated before $y$. Let $t$ be the time when $y$ has been activated. Then by the activation rule, $\mathcal{D}(x,y) > r_t(x)$ and by Lemma 8, $r_t(x) > \frac{1}{6}\Delta(x) \geq \frac{1}{6}\min(\Delta(x), \Delta(y))$. $\qquad\square$

**Corollary 10.** *For each arm $x$, we have $v_T(x) \leq O(\log \frac{T}{\delta})\Delta^{-2}(x)$.*

*Proof.* Use Lemma 8 for $t = T$ and plug in the definition of the confidence radius $r_t(x)$. $\qquad\square$

## A.2 PROOF OF THEOREM 1

We follow the proof idea of Kleinberg et al. (2019) and Joulani et al. (2013). Fix round $t$, let $V_t$ be the set of all arms that are active at time $t$, and let

$$A_{(i,t)} = \left\{x \in V_t : 2^i \leq \frac{1}{\Delta(x)} < 2^{i+1}\right\} = \left\{x \in V_t : 2^{-i-1} < \Delta(x) \leq 2^{-i}\right\}.$$

Recall that by Corollary 10 for each $x \in A_{(i,t)}$, we have $v_t(x) \leq O(\log \frac{t}{\delta})\Delta^{-2}(x)$. Therefore,

$$\sum_{x \in A_{(i,t)}} \Delta(x)v_t(x) \leq O(\log \frac{t}{\delta}) \sum_{x \in A_{(i,t)}} \frac{1}{\Delta(x)} \leq O(2^i \log \frac{t}{\delta})|A_{(i,t)}|.$$

Let $r_i = 2^{-i}$, note that by Corollary 9, any set of radius less than $r_i/16$ contains at most one arm from $A_{(i,t)}$. It follows that $|A_{(i,t)}| \leq N_z(r_i) \leq cr_i^{-d_z}$. Hence,

$$\sum_{x \in A_{(i,t)}} \Delta(x)v_t(x) \leq O(\log \frac{t}{\delta})\frac{1}{r_i} \cdot N_z(r_i) \leq O(\log \frac{t}{\delta})cr_i^{-d_z-1}.$$

For any $\rho \in (0,1)$, we have

$$\begin{aligned}
\sum_{x \in V_t} \Delta(x)n_t(x) &= \sum_{x \in V_t : \Delta(x) \leq \rho} \Delta(x)n_t(x) + \sum_{x \in V_t : \Delta(x) > \rho} \Delta(x)n_t(x) \\
&\leq \rho t + \sum_{x \in V_t : \Delta(x) > \rho} \Delta(x)(v_t(x) + w_t(x)) \\
&\leq \rho t + \sum_{i < \log_2(1/\rho)} \sum_{x \in A_{(i,t)}} \Delta(x)(v_t(x) + w_t^*(x)),
\end{aligned} \qquad (4)$$

where $w_t^*(x) = \max_{s \leq t} w_s(x)$. For the first part of the sum sequence,

$$\sum_{i < \log_2(1/\rho)} \sum_{x \in A_{(i,t)}} \Delta(x)v_t(x) \leq \sum_{i < \log_2(1/\rho)} O(\log \frac{t}{\delta})cr_i^{-d_z-1} \leq O(c\log \frac{t}{\delta})\left(\frac{1}{\rho}\right)^{d_z+1}.$$

Since the delay is bounded, we have $w_t^*(x) \leq \tau_{\max}$, and hence

$$\sum_{i < \log_2(1/\rho)} \sum_{x \in A_{(i,t)}} \Delta(x)w_t^*(x) \leq \sum_{i < \log_2(1/\rho)} \tau_{\max} \cdot cr_i^{-d_z} \leq \tau_{\max} \cdot c\left(\frac{1}{\rho}\right)^{d_z}.$$

Therefore, take expectation on both sides of Eq 4 and set $t = T$, we have

$$R(T) \lesssim \rho T + c\log \frac{T}{\delta}\left(\frac{1}{\rho}\right)^{d_z+1} + \tau_{\max} \cdot c\left(\frac{1}{\rho}\right)^{d_z},$$

where $\lesssim$ denotes "less in order". Since it holds for any $\rho \in (0,1)$, hence by taking $\rho = \left(\frac{\log T}{T}\right)^{\frac{1}{d_z+2}}$, we have

$$R(T) \leq O\left(T^{\frac{d_z+1}{d_z+2}}\left(c\log \frac{T}{\delta}\right)^{\frac{1}{d_z+2}} + c\tau_{\max} \cdot \left(\frac{T}{\log T}\right)^{\frac{d_z}{d_z+2}}\right) = \tilde{O}\left(T^{\frac{d_z+1}{d_z+2}} + \tau_{\max}T^{\frac{d_z}{d_z+2}}\right).$$

This completes the proof. $\qquad\square$

# B ANALYSIS OF DELAYED LIPSCHITZ PHASED PRUNING

*Proof sketch.*

- With high probability, for any arm $x \in B$ and any surviving active ball $B$ in of phase $m$, the empirical average reward $\hat{\mu}_m(B)$ concentrates around the underlying $\mu(x)$. This concentration result can guarantee that regions with unfavorable rewards will be adaptively eliminated. Denote this event as $\mathcal{E}_1$.

- With high probability, $n_t(B)$ can be upper bounded by $v_{t+Q(p)}(B)$ with some factor of quantile $p$. This connection allows us to upper bound the pseduo regret. Denote this event as $\mathcal{E}_2$.

- Under event $\mathcal{E} = \mathcal{E}_1 \cap \mathcal{E}_2$, the optimal arm $x^* = \arg\max_{x \in \mathcal{A}} \mu(x)$ is not eliminated after phase $m$. This shows that the optimal arm survives all eliminations with high probability thus we are pruning the unfavorable regions correctly.

- Under event $\mathcal{E}$, the suboptimality gap is bounded by the radius, i.e. $\Delta(x) \leq 16 r_{m-1}$, thus the surviving active balls are those with promising rewards.

## B.1 SOME USEFUL LEMMAS

**Lemma 11.** *For phase $m$ that is complete (entered Pruning, line 13 of algorithm 2), define $t_m$ as the last round of it and let $t_0 = 0$, clearly $t_m \leq T$. Let $m^*$ be the last complete phase such that $t_{m^*} \leq T$. Define the following events:*

$$\mathcal{E}_1 := \left\{ |\hat{\mu}_m(B) - \mu(x)| \leq 2r_m + \sqrt{\frac{4\log T + 2\log(2/\delta)}{v_m}}, \forall x \in B, \forall B \in \mathcal{B}_m, \forall 1 \leq m \leq m^* \right\},$$

*where $\hat{\mu}_m(B)$ is calculated when $B$ is removed from $\mathcal{B}_m^+$. It holds that $\Pr[\mathcal{E}_1] \geq 1 - \delta$.*

*Proof.* For a fixed ball $B \in \mathcal{B}_m$ that is removed from $\mathcal{B}_m^+$, there are $v_m = (2\log T + \log(2/\delta))/2r_m^2$ observations from $B$ has been observed, thus the empirical average reward of $B$ and its expectation is

$$\hat{\mu}_m(B) = \frac{1}{v_m} \sum_{i=1}^{v_m} y_{B,i}, \quad \text{and} \quad \mathbb{E}[\hat{\mu}_m(B)] = \frac{1}{v_m} \sum_{i=1}^{v_m} \mu(x_{B,i}),$$

where $(x_{B,i}, y_{B,i})$ denote the $i$-th arm-reward pair from ball $B$.

By the assumption of white noise, $\hat{\mu}_m(B) - \mathbb{E}[\hat{\mu}_m(B)]$ is sub-Gaussian with parameter $\sqrt{\frac{1}{v_m}}$. Hence by Hoeffding inequality,

$$\Pr\left( |\hat{\mu}_m(B) - \mathbb{E}[\hat{\mu}_m(B)]| \geq \sqrt{\frac{4\log T + 2\log(2/\delta)}{v_m}} \right)$$

$$\leq 2 \cdot \exp\left( -\frac{v_m}{2} \cdot \frac{4\log T + 2\log(2/\delta)}{v_m} \right) \quad \leq \frac{\delta}{T^2}.$$

By Lipschitzness condition, for any $x \in B$ we have

$$|\mathbb{E}[\hat{\mu}_m(B)] - \mu(x)| = \left| \frac{1}{v_m} \sum_{i=1}^{v_m} (\mu(x_{B,i}) - \mu(x)) \right| \leq 2r_m.$$

Therefore,

$$\Pr\left( |\hat{\mu}_m(B) - \mu(x)| \leq 2r_m + \sqrt{\frac{4\log T + 2\log(2/\delta)}{v_m}}, \quad \forall x \in B \right) \geq 1 - \frac{\delta}{T^2}.$$

Now for any $\mathcal{B}_m$ that a phase $m$ is complete, each ball $B \in \mathcal{B}_m$ is played at least once, thus $|\mathcal{B}_m| \leq T$. Take the union bound over all $B \in \mathcal{B}_m$ yields

$$\Pr\left( |\hat{\mu}_m(B) - \mu(x)| \leq 2r_m + \sqrt{\frac{4\log T + 2\log(2/\delta)}{v_m}}, \quad \forall x \in B, \forall B \in \mathcal{B}_m \right) \geq 1 - \frac{\delta}{T}.$$

Now since the number of finished phase is no more than time horizon $T$, take another union bound over all finished phase $1 \leq m \leq m^*$ yields

$$\Pr[\mathcal{E}_1] \geq 1 - \delta,$$

as desired. $\qquad \square$

**Lemma 12.** *Suppose time round $t$ is in phase $m$. For any ball $B \in \mathcal{B}_m$ and quantile $p \in (0, 1]$, we have*

$$\Pr\left(v_{t+Q(p)}(B) \leq \frac{p}{2}n_t(B)\right) \leq \exp\left(-\frac{p}{8}n_t(B)\right),$$

*where $v_t(B)$ denote the number of times of rewards from ball $B$ observed at the end of round $t - 1$, and $n_t(B)$ denote the number of times the agent plays ball $B$.*

*Proof.* Let $\mathbb{I}\{\cdot\}$ denote the indicator function. By definiton of quantile funciton, $\mathbb{E}[\mathbb{I}\{\tau_s \leq Q(p)\}] = \Pr[\tau_s \leq Q(p)] \geq p$. Let $S_t(B) := \{s \mid t_{m-1} < s \leq t, x_s \in B\}$ be a set of time round for which the agent plays ball $B$, note that $n_t(B) = |S_t(B)|$, and thus we have

$$\Pr\left(v_{t+Q(p)}(B) \leq \frac{p}{2}n_t(B)\right) \leq \Pr\left(\sum_{s \in S_t(B)} \mathbb{I}\{s + \tau_s \leq t + Q(p)\} \leq \frac{p}{2}n_t(B)\right)$$

$$\leq \Pr\left(\sum_{s \in S_t(B)} \mathbb{I}\{\tau_s \leq Q(p)\} \leq \frac{p}{2}n_t(B)\right)$$

$$\leq \Pr\left(\sum_{s \in S_t(B)} \mathbb{I}\{\tau_s \leq Q(p)\} \leq \frac{1}{2}\sum_{s \in S_t(B)} \mathbb{E}[\mathbb{I}\{\tau_s \leq Q(p)\}]\right)$$

$$\stackrel{(i)}{\leq} \exp\left(-\frac{1}{8}\sum_{s \in S_t(B)} \mathbb{E}[\mathbb{I}\{\tau_s \leq Q(p)\}]\right)$$

$$\leq \exp\left(-\frac{p}{8}n_t(B)\right),$$

where the inequality (i) follows from the multiplicative Chernoff bound. $\qquad \square$

**Lemma 13.** *Define the following event:*

$$\mathcal{E}_2 := \left\{n_t(B) < \frac{24\log T + 8\log(1/\delta)}{p} \quad or \quad v_{t+Q(p)}(B) \geq \frac{p}{2}n_t(B),\right.$$

$$\left. \forall t \in S_{t_m}(B), \forall B \in \mathcal{B}_m, \forall 1 \leq m \leq m^*\right\}$$

*It holds that $\Pr[\mathcal{E}_2] \geq 1 - \delta$.*

*Proof.* It suffices to show that for the complementary event, $\Pr[\mathcal{E}_2^c] \leq \delta$. By Lemma 12 and applying union bound, we have

$$\Pr[\mathcal{E}_2^c] = \Pr\left[\exists m, B \in \mathcal{B}_m, t \in S_{t_m}(B):\right.$$

$$\left. n_t(B) \geq \frac{24\log T + 8\log(1/\delta)}{p}, v_{t+Q(p)}(B) < \frac{p}{2}n_t(B)\right]$$

$$\leq \sum_m \sum_{B \in \mathcal{B}_m} \sum_{\substack{t \in S_{t_m}(B), \\ n_t(B) \geq \frac{24\log T + 8\log(1/\delta)}{p}}} \Pr\left(v_{t+Q(p)}(B) < \frac{p}{2}n_t(B)\right)$$

$$\leq \sum_m \sum_{B \in \mathcal{B}_m} \sum_{\substack{t \in S_{t_m}(B), \\ n_t(B) \geq \frac{24\log T + 8\log(1/\delta)}{p}}} \exp\left(-\frac{p}{8}n_t(B)\right)$$

$$\overset{(ii)}{\leq} T^3 \cdot \exp\left(-\frac{p}{8} \cdot \frac{24\log T + 8\log(1/\delta)}{p}\right) \quad \leq \delta,$$

where the inequality (ii) uses the union bound trick in Lemma 11. This concludes the proof. $\square$

**Corollary 14.** *Define the clean event $\mathcal{E} = \mathcal{E}_1 \cap \mathcal{E}_2$. By union bound, it holds that $\Pr[\mathcal{E}] \geq 1 - 2\delta$.*

**Lemma 15.** *Under event $\mathcal{E}$, the optimal arm $x^* = \arg\max \mu(x)$ is not pruned after the first $m^*$ phases.*

*Proof.* Let $B^* \in \mathcal{B}_m$ denote the ball that contains $x^*$ in phase $m$. It suffices to show that $B^*$ is not pruned. Under event $\mathcal{E}_1$, for any ball $B \in \mathcal{B}_m$ and $x \in B$, we have

$$\hat{\mu}_m(B) - \mu(x) \leq 2r_m + \sqrt{\frac{4\log T + 2\log(2/\delta)}{v_m}},$$

and

$$\mu(x^*) - \hat{\mu}_m(B^*) \leq 2r_m + \sqrt{\frac{4\log T + 2\log(2/\delta)}{v_m}}.$$

Now, use the definition of $v_m$ and the fact that $\mu(x^*) \geq \mu(x)$, taking the sum of the previous two inequality yields

$$\hat{\mu}_m(B) - \hat{\mu}_m(B^*) \leq 8r_m.$$

Therefore, by the pruning rule, $B^*$ is not pruned. $\square$

**Lemma 16.** *Under event $\mathcal{E}$, for any phase $1 \leq m \leq m^* + 1$, any $B \in \mathcal{B}_m$ and any $x \in B$, it holds that*

$$\Delta(x) \leq 16r_{m-1}.$$

*Proof.* For $m = 1$, it hold trivially that $16r_{m-1} = 16 > 1 \geq \Delta(x)$. For $2 \leq m \leq m^* + 1$, we consider the previous complete phase $m - 1$ so that the lemma also holds for the incomplete phase $m^* + 1$. Suppose for any ball $B \in \mathcal{B}_m$ and $x \in B, x^* \in B_m^*$, in phase $m - 1$, $x \in B_0 \in \mathcal{B}_{m-1}$, where $B_0$ is the parent ball of $B$ such that it contains $x$ in phase $m - 1$, and let $x^* \in B_{m-1}^* \in \mathcal{B}_{m-1}$ since the optimal arm is not pruned. By Lemma 11, for any $B \in \mathcal{B}_m$ and any $x \in B$

$$\Delta(x) = \mu^* - \mu(x) \leq \hat{\mu}_{m-1}(B_{m-1}^*) - \hat{\mu}_{m-1}(B_0) + 4r_{m-1} + 2\sqrt{\frac{4\log T + 2\log(2/\delta)}{v_{m-1}}}.$$

Now, use the definition of $v_{m-1}$, we have

$$\Delta(x) \leq \hat{\mu}_{m-1}(B_{m-1}^*) - \hat{\mu}_{m-1}(B_0) + 8r_{m-1}.$$

By pruning rule, since $B_0$ is not pruned, we have

$$\hat{\mu}_{m-1}(B_{m-1}^*) - \hat{\mu}_{m-1}(B_0) \leq \hat{\mu}_{m-1}^* - \hat{\mu}_{m-1}(B_0) \leq 8r_{m-1}.$$

It follows that $\Delta(x) \leq 16r_{m-1}$. $\square$

## B.2 PROOF OF THEOREM 3

We modify the proof in Feng et al. (2022) by relating $n_m$ and $v_m$ using previous lemmas to show that the addtional term in the regret bound incurred by the delays scales with quantiles.

*Proof.* Fix $p \in (0,1]$, and define $\mathcal{E} = \mathcal{E}_1 \cap \mathcal{E}_2$ as in Lemma 12, 13, it holds that $\Pr[\mathcal{E}] \geq 1 - 2\delta$. For any phase $m$ and any ball $B \in \mathcal{B}_m$, let $s_B$ be the last time that ball $B$ is played, thus we have $v_{s_B-1}(B) < v_m = (2\log T + \log(2/\delta))/2r_m^2$. Under the clean event $\mathcal{E}_2$, at least one of the two following event is true:

$$n_{s_B-Q(p)-1}(B) \leq \frac{2}{p}v_{s_B-1}(B) \leq \frac{2\log T + \log(2/\delta)}{pr_m^2},$$

or

$$n_{s_B-Q(p)-1}(B) \leq \frac{24\log T + 8\log(1/\delta)}{p} \leq \frac{24\log T + 8\log(2/\delta)}{pr_m^2}.$$

Therefore, the total number ball $B$ is played can be bounded as

$$n_{s_B}(B) = n_{s_B-Q(p)-1}(B) + (n_{s_B}(B) - n_{s_B-Q(p)-1}(B))$$
$$\leq \frac{24\log T + 8\log(2/\delta)}{pr_m^2} + \sum_{t\in[s_B-Q(p),s_B]} \mathbb{I}\{x_t \in B\}$$

Let $u_m(B)$ be the number of balls in the round-robin process of phase $m$ at time $\min\{s_B, t_m\}$, since the algorithm runs over $u_m(B)$ balls in a round-robin fashion, it follows that

$$\sum_{t\in[s_B-Q(p),s_B]} \mathbb{I}\{x_t \in B\} \leq \frac{Q(p)+1}{u_m(B)}.$$

Also, we have that

$$\sum_{B\in\mathcal{B}_m} \frac{1}{u_m(B)} \leq \frac{1}{|\mathcal{B}_m|} + \frac{1}{|\mathcal{B}_m|-1} + \cdot\cdot + \frac{1}{2} + 1 \leq \log|\mathcal{B}_m| + 1.$$

Now, fix the total number of phases $M$. Any arm played after phase $M$ attains a regret bounded by $16r_M$, since the balls played after phase $M$ have radius no larger than $r_M$. By Lemma 16 and definition of zooming number, it follows that

$$|\mathcal{B}_m| \leq N_z(r_m) \leq cr_m^{-d_z} \leq c\cdot 2^{d_z m}.$$

The regret $R(T)$ can be bounded by

$$R(T) \leq 16r_M T + \sum_{m=1}^{M} \sum_{B\in\mathcal{B}_m} \sum_{i=1}^{n_{s_B}(B)} \Delta(x_{B,i})$$

$$\leq 16r_M T + 16\sum_{m=1}^{M} \sum_{B\in\mathcal{B}_m} r_m \cdot n_{s_B}(B)$$

$$\leq 16r_M T + 16\sum_{m=1}^{M} \sum_{B\in\mathcal{B}_m} \left[\frac{24\log T + 8\log(2/\delta)}{pr_m} + r_m \sum_{t\in[s_B-Q(p),s_B]} \mathbb{I}\{x_t \in B\}\right]$$

$$\leq 16r_M T + 16\sum_{m=1}^{M} \left[|\mathcal{B}_m| \cdot \frac{24\log T + 8\log(2/\delta)}{p\cdot 2^{-m}} + 2^{-m}(Q(p)+1)(\log|\mathcal{B}_m|+1)\right]$$

$$\leq 16r_M T + 16\sum_{m=1}^{M} \left[c\cdot 2^{(d_z+1)m} \cdot \frac{24\log T + 8\log(2/\delta)}{p} + 2^{-m}(Q(p)+1)(d_z m+1)\right]$$

$$\leq 16\cdot 2^{-M}\cdot T + 32\cdot 2^{(d_z+1)M}\cdot c\cdot \frac{24\log T + 8\log(2/\delta)}{p} + 16(Q(p)+1)(3d_z+1)$$

$$\lesssim 2^{-M}\cdot T + 2^{(d_z+1)M}\cdot c\cdot \frac{\log(T/\delta)}{p} + Q(p),$$

where $\lesssim$ denotes "less in order".

Since the inequality holds for any $M$, hence by taking $M = \frac{\log\frac{T}{c\log T}}{d_z+2}$, we have

$$R(T) \lesssim \frac{1}{p}T^{\frac{d_z+1}{d_z+2}} \left(c\log\frac{T}{\delta}\right)^{\frac{1}{d_z+2}} + Q(p).$$

This holds for any $p \in (0, 1]$, thus we further minimize over $p$ to obtain the desired lowest possible upper bound as stated in the theorem. $\qquad\square$

## C  ANALYSIS OF LOWER BOUND

We will use the following lower bound for Lipschitz bandtis from Theorem 7 of Slivkins (2011).

**Theorem 17.** *Consider the Lipschitz Bandit problem on a metric space $(\mathcal{A}, \mathcal{D})$. Define*

$$R_c(T) = C_0 \inf_{r_0 \in (0,1)} \left( r_0 T + \log T \sum_{r=2^{-i}: i \in \mathbb{N}, r_0 \leq r \leq 1} \frac{1}{r} N_c(r) \right)$$

*where $N_c(r)$ is the $r$-covering number of $(\mathcal{A}, \mathcal{D})$, and $C_0 = O(1)$. Fix a time horizon $T$ and a positive number $R \leq R_c(T)$, then there exists a distribution $\mathcal{I}$ over problem instances on $(\mathcal{A}, \mathcal{D})$ such that*

(a) *For each problem instance $I \in \mathcal{I}$, we have $R_z(T) \leq O(R)$, where*

$$R_z(T) = C_0 \inf_{r_0 \in (0,1)} \left( r_0 T + \log T \sum_{r=2^{-i}: i \in \mathbb{N}, r_0 \leq r \leq 1} \frac{1}{r} N_z(r) \right),$$

*and $N_z(r)$ is the $r$-zooming number of $(\mathcal{A}, \mathcal{D}, \mu)$, and $C_0 = O(1)$.*

(b) *For any algorithm $\mathcal{M}$, there exists at least one problem instance $I \in \mathcal{I}$ on which the expected regret of $\mathcal{M}$ satisfies $R(T) \geq \Omega(R/\log T)$.*

Now we will adapt this lower bound to a delay-presence version using a reduction techinique introduced in Lancewicki et al. (2021).

*proof of Theorem 5.* We consider a specified delay distribution such that the delay is 0 with probability $p$, and $\infty$ otherwise (missing feedback). Let $\mathcal{M}$ be any delayed Lipschitz bandits algorithm, and we use the reduction techinique to build an algorithm $\mathcal{M}_0$ that simulates $\mathcal{M}$ and interacts with a non-delayed environment for $\frac{pT}{4}$ rounds. In each round $t$, a Bernoulli variable $Z_t$ with success probability $p$ is sampled. If $Z_t = 1$, $\mathcal{M}_0$ choose the same action as $\mathcal{M}$ and receive its feedback. Otherwise, only $\mathcal{M}$ plays this round and skip $\mathcal{M}_0$. It is with high probability that $\mathcal{M}_0$ has played for $\frac{pT}{4}$ rounds after $(1 - p/4)T$ rounds, otherwise for the rest of the round let $\mathcal{M}_0$ follow $\mathcal{M}$'s actions and receive feedback with probability $p$. Define the failure event as

$$F = \left\{ \sum_{t<(1-p/4)T} Z_t < \frac{pT}{4} \right\}.$$

Since $(1 - p/4)T \leq T/2$, By the multiplicative Chernoff bound,

$$\Pr[F] \leq \Pr\left\{ \sum_{t<=T/2} Z_t < \frac{pT}{4} \right\} \leq \exp\left( -\frac{pT}{16} \right).$$

By Theorem 17, there exists a universal constant $C_0$ such that for any Lipschitz Bandit algorithm $\mathcal{M}_0$,

$$R_{\mathcal{M}_0}(T) \geq \frac{C_0 R}{\log T}.$$

On the other hand, let $V_T$ be the active set of $\mathcal{M}$, it must contains the active set of $\mathcal{M}_0$. Therefore,

$$
\begin{aligned}
R_{\mathcal{M}_0}\left( \frac{1}{4}pT \right) &\leq \mathbb{E}\left[ \sum_{t<(1-p/4)T} \mathbb{I}\{Z_t = 1\} \sum_{x \in V_T} \mathbb{I}\{x_t = x\}\Delta(x) \right] \\
&\quad + \mathbb{E}\left[ \mathbb{I}\{F\} \sum_{t \geq (1-p/4)T} \sum_{x \in V_T} \mathbb{I}\{x_t = x\}\Delta(x) \right] \\
&\leq \sum_{t<(1-p/4)T} \mathbb{E}[\mathbb{I}\{Z_t = 1\}] \mathbb{E}\left[ \sum_{x \in V_T} \mathbb{I}\{x_t = x\}\Delta(x) \right] + \frac{pT}{4}\Pr[F] \\
&\leq p\mathbb{E}\left[ \sum_{t=1}^{T} \sum_{x \in V_T} \mathbb{I}\{x_t = x\}\Delta(x) \right] + \frac{pT}{4}\exp\left( -\frac{pT}{16} \right) \leq pR_{\mathcal{M}}(T) + 4
\end{aligned}
$$

Therefore,

$$R_{\mathcal{M}}(T) \geq \frac{C_0 R}{p \log\left(\frac{1}{4}pT\right)} - \frac{4}{p} \tag{5}$$

Consider a specified delay distribution such that the delay is a fixed value $\tau_0$ with probability $p$, and $\infty$ otherwise (missing feedback). Under this delay distribution, the algorithm does not get any feedback in the first $\tau_0 = Q(p)$ rounds, and the agent is unable to distinguish between arms in this initial period, so any algorithm in this period is no better than a uniform random play on $\mathcal{A}$. Therefore, for a fixed problem instance $(\mathcal{A}, \mathcal{D}, \mu)$ (and hence the suboptimality gaps are fixed), the expected regret of any algorithm will in this initial period be at least $\tau_0 \bar{\Delta}$, where $\bar{\Delta} = \int_{\mathcal{A}} \Delta(x)/\int_{\mathcal{A}} 1$ is the average suboptimality gaps on $\mathcal{A}$, denotes the average regret. The regret in this initial period, along with Eq 5, yields

$$R_{\mathcal{M}}(T) \gtrsim \frac{R}{p \log T} - \frac{1}{p} + \bar{\Delta} \cdot Q(p).$$

This completes the proof. $\qquad\square$

## D  ADDITIONAL EXPERIMENTAL DETAILS

In the analysis and algorithms of our main paper we assume the sub-Gaussian parameter of white noise is $\sigma = 1$. In reality, if the value or an upper bound of $\sigma$ is known or can be estimated, we could easily modify the components in our proposed algorithms.

- Delayed Zooming Algorithm 1: we can modify the confidence radius by multiplying $\sigma$:

$$r_t(x) = \sigma \sqrt{\frac{4 \log T + 2 \log(2/\delta)}{1 + v_t(x)}}.$$

- Delayed Lipschitz Phased Pruning 2: we can modify the required number of observations for each ball in each phase by multiplying $\sigma^2$:

$$v_m = \sigma^2 \cdot \frac{2 \log T + \log(2/\delta)}{2 r_m^2}.$$

Indeed, a larger sub-Gaussian parameter indicates greater variability in the noise distribution, leading to increased uncertainty in the reward estimates. Consequently, the algorithms must either enlarge the confidence radius or increase the number of observations required for each ball in each phase to ensure sufficient exploration under higher noise conditions.

The numerical results of the final cumulative regrets (at $T = 60000$) in our simulations in Section 7 (Figure 1) are displayed in Table 1.

## E  THE USE OF LARGE LANGUAGE MODELS (LLMS)

We used the Large Language Model (LLM) sparingly for minor writing tasks such as grammar and spell-checking. All research, deduction, data analysis, and the core content of this paper were developed independently by the authors without assistance from an LLM.

| | Algorithm | $\mathbb{E}[\tau]$ | Uniform | Geometric |
|---|---|---|---|---|
| | | 0 | 138.97 | 138.97 |
| | Delayed Zooming | 20 | 154.55 | 159.30 |
| Triangle reward function | | 50 | 171.07 | 152.98 |
| | | 0 | 304.60 | 304.60 |
| | DLPP | 20 | 314.87 | 312.44 |
| | | 50 | 326.71 | 325.74 |
| | Algorithm | $\mathbb{E}[\tau]$ | Uniform | Geometric |
| | | 0 | 130.64 | 130.64 |
| | Delayed Zooming | 20 | 137.31 | 132.88 |
| Sine reward function | | 50 | 148.69 | 144.08 |
| | | 0 | 178.05 | 178.05 |
| | DLPP | 20 | 195.35 | 186.28 |
| | | 50 | 209.97 | 208.80 |
| | Algorithm | $\mathbb{E}[\tau]$ | Uniform | Geometric |
| | | 0 | 1445.86 | 1445.86 |
| | Delayed Zooming | 20 | 1843.05 | 1463.38 |
| Two dim reward function | | 50 | 1858.45 | 1828.15 |
| | | 0 | 1120.64 | 1120.64 |
| | DLPP | 20 | 1159.85 | 1120.63 |
| | | 50 | 1136.46 | 1142.55 |

Table 1: Numerical values of final cumulative regrets of different algorithms under the experimental settings used in Figure 1 in Section 7

