# OpenReview forum: "Lipschitz Bandits with Stochastic Delayed Feedback"
_ICLR.cc/2026/Conference — ICLR 2026 Poster_

### Official Review · Reviewer_ysrv · 2025-10-29

**Soundness:** 4
**Presentation:** 3
**Contribution:** 3
**Rating:** 8
**Confidence:** 3

**Summary:**

This paper studies Lipschitz bandits (that is, the action set is continuous defined over a metric space) with stochastic delayed feedback. The authors give algorithms for both bounded and unbounde delay setting with theoretical bounds.

There's also a lower bound given to demonstrate the nearly tightness of upper bounds. There are simulation results as well but given the literature there's no baseline algs. to compare with.

**Strengths:**

1. A new setup that hasn't been studied and requires new design in the algorithm.
2. Lower bound compliments the results

**Weaknesses:**

n/a.

**Questions:**

1. My understanding from Sec. 4 is that, the challenge comes from the correlation between arms (observing one arm also helps learn its neighbors), which indeed prevents us from showing some certain ``suboptimality gap inequality.'' However, intuitively, wouldn't the shrinking confidence level a good thing, which means the uncertainty is reduced faster? Why does it turn out to be an ``issue'' rather than something we can/should leverage?
2. Could the authors elaborate more on the computational complexity of Alg. 2?
3. Even in the bounded delay setting, ``This implies that feedback is always eventually observed and never missing,'' is not precise, because feedback can still come after $t=T$?
4. Some minor issues: ``BOLD and QPM-D'' appears in Line 094, but it's hard to understand what they are; $d_z$ appears in Line 063 without even an informal definition.

---

> ### Author Response · Authors · 2025-11-19
>
> (Q1) We thank the reviewer for this excellent question. We agree that a shrinking confidence radius, which signifies reduced uncertainty, is fundamentally a good outcome that should ideally be leveraged. The core challenge in our analysis under delays is that the certainty reduction is decoupled in time and location, which complicates the proof structure required for regret bounds. In the delay-free setting, the confidence radius of the pulled arm shrinks immediately and locally at the time of the pull, providing a clean coupling between action and information gain. However, when delay is introduced, this coupling breaks down in two ways: 1. Temporal Decoupling: The shrinkage is postponed from the time the arm is pulled ($t$) to the time the reward is observed ($t+\tau$). Therefore, the uncertainty is reduced slower, not faster, from the perspective of the current round. 2. Action Decoupling: The arm whose confidence radius shrinks at time $t$ (when an old reward arrives) may not be the arm pulled at time $t$. This asynchronous information update prevents the use of standard analysis techniques that rely on the assumption that an action at $t$ leads to a direct reduction in the uncertainty of that action set at $t+1$. This loss of a direct temporal and local relationship between pull and information gain is the main reason the analysis becomes challenging.
>
> (Q2) At each round $t$, only one arm is pulled, which triggers at most one update to $\hat{\mu}_m(C)$, depending on whether the corresponding feedback has arrived. The value $\hat{\mu}_m^*$ can also be updated whenever $\hat{\mu}_m(C)$ is updated. Each of these operations takes $O(1)$ time, so the total running time is $O(T)$. The $r_m$-covering can be precomputed in advance, and its computational complexity depends on the specific metric space, as well as the oracle. Note this discretization part is unavoidable for any Lipschitz bandit algorithms.
>
> (Q3) Thank you for bringing this to our attention. As you note, feedback is observed for all rounds prior to $t = T - \tau_{\max}$, and it can only be absent in the final $\tau_{\max}$ rounds. We have updated the revision accordingly.
>
> (Q4) Thank you for raising these issues. We have elaborated on the discussion of BOLD (Black-box Online Learning under Delayed feedback) and QPM-D (Queued Partial Monitoring with Delays), and included the definition of $d_z$ in the revision.

---

### Official Review · Reviewer_2LpJ · 2025-11-01

**Soundness:** 3
**Presentation:** 2
**Contribution:** 3
**Rating:** 6
**Confidence:** 4

**Summary:**

The work studies the problem of Lipschitz bandits problem where the reward is not observed immediately rather after a random delay. The authors develop two algorithms, a zooming algorithm called as Delayed Zooming Algorithm for bounded delays and DLPP for unbounded delay. They also showcase the optimal performance of these algorithm with theoretical guarantees in regret bound. To support this regret guarantee, they also provide lower bound guarantee. The theoretical work is complemented with the experimentation to show the effectiveness of the algorithm.

**Strengths:**

The problem formulation is new and interesting with Lipschitz bandits having stochastic delayed feedback.

The work proposed two algorithms covering two regimes wrt delay, Delayed Zooming Algorithm for bounded delay and DLPP for unbounded delay.

The lazy update trick is neat wrt confidence radius on unpulled arm.

The work also includes matching lower bound that aligns with the upper bound up to log factor and is important to showcase the importance of the upper bound

**Weaknesses:**

Alg 1 and Alg 2 both assume that they have access to a oracle for covering but it is not the case in many practical scenarios and can be expensive.

Both the delay and reward distribution are assumed to be independent and it is not the case in many real world system.

The comparisons are not established with respect to other censored bandit setting or closer setting that could be made to work with some relaxation for considering them for baseline. This could have improved the experimentation setting to help understand the performance gains.

Also, the experimental setting is minimal with only simulations having different delay distributions. There is no real world dataset used to evaluate the experiments.

**Questions:**

Since Algorithm 1 requires bounded delay \tau_{max}, can delay from the distribution be scaled down to account of practical situation (in case a wide delay Random variable has to be accommodated) so the regret guarantees still holds ?

The problem setting assumes that the delay distribution and reward distribution are independent. However, in many situation like online advertisement where the delay (conversion) in getting a reward is often associated with reward. In this case, how would the analysis break if the delay depends on the reward?

In DLPP, phases end after every ball gets $v_{m}$ samples to be observed and since $\tau$ also includes $\inf$, and if the prob. distribution is supported more on the $\inf$ (so reward is never observed) so how does the algorithm overcome this scenario.

You measure regret on generated rewards not only on the observed reward. In real systems we only evaluate on observed conversions. How different would the bounds look under only observed reward and would the algorithm need to change to accommodate this ?

Also, if $P(\tau = \inf) > 0 on delay distribution, Does that not reduce to the standard Censored Bandits ? If it does reduce, how does the regret of proposed algorithm compare against the Censored Bandits.

---

> ### Author Response · Authors · 2025-11-19
>
> (Weakness 1) We thank the reviewer for raising the practical cost of the covering oracle. We acknowledge that constructing a fine covering can be expensive in practice. However, the assumption of access to such an oracle is common and necessary within the theoretical framework of Lipschitz bandit problems (Kleinberg et al., 2019). Our paper's primary focus is on the regret analysis of the two proposed algorithms. Exploring computationally cheap and adaptive covering strategies is a highly valuable extension we aim to explore in future work.
>
> (Weakness 2 and Q2) We agree that reward-dependent delays scenarios pose a crucial real world challenge. As the first work to study the Lipschitz bandits with stochastic delays, we began with the standard independent reward and delay assumption (e.g. Vernade et al., 2018). Our Delayed Zooming algorithm's core analysis remains valid even with this dependence, provided the maximal delay ($\tau_{\max}$) is bounded. This bound ensures for sufficient rewards eventually arrive to determine if an arm is promising. However, DLPP analysis does break down under this dependence, as the empirical mean rewards become biased estimators (e.g., if lower rewards have shorter delays), skewing the estimation of active regions. We believe designing algorithms work for this more interesting and complex setting is a promising direction of future work.
>
> (Weakness 3) We appreciate the suggestion regarding broader experimental comparison. However, our work addresses the Lipschitz Bandits with stochastic delays, which is a highly specialized line of work. To the best of our knowledge, we are the first to design algorithms and establish regret bounds in this setting. Existing algorithms for delayed feedback typically operate under the standard MAB or LinUCB settings, making their fundamental assumptions incompatible with the metric structure required by Lipschitz bandits. Modifying these baselines to fit the continuous arm space and satisfy the Lipschitz constraints while preserving their original theoretical guarantees is a non-trivial research endeavor that goes beyond the scope of this work. Our current simulation result, which includes the delay free setting, validates our approach against a proximal challenge.
>
> (Weakness 4) We acknowledge the desire for real-world validation. However, the core contribution of this paper is the theoretical analysis and the derivation of the first established regret bounds for Lipschitz Bandits with stochastic delays. Obtaining reliable real-world datasets with a continuous arm space that satisfies the Lipschitz constraints and precisely measured stochastic delay is highly challenging. Consequently, all existing research on Lipschitz bandits relies exclusively on simulation studies, as no suitable real-world datasets currently exist.  We view the pursuit of real-world validation as a key step for future work.
>
> (Q1) Thank you for this interesting practical question. For a wide delay random variable (the delay can be very large), by theorem 1 (line 262), as long as the maximal delay $\tau_{\max}$ only scales with $\log T$, the regret bound still holds. To scale down $\tau_{\max}$, if one choose to cap the delay at a smaller, fixed $\tau_{\max}' < \tau_{\max}$, any true rewards arriving after that threshold are effectively censored (lost), this does not fit our setting anymore, so the regret bound does not hold if choose to censored at a smaller threshold for delays. However, one possible modification is that one can use an estimated mean reward to replace the censored reward, then the regret bound remains valid. We believe this can be an interesting extension in future work.
>
> (Q3) When the distribution places more mass near infinity, DLPP indeed requires more time to accumulate sufficient samples for each phase. Nevertheless, it is still able to correctly identify the more promising region as the algorithm progresses. For a fixed $p$, the quantile $Q(p)$ becomes larger under such distributions, which in turn leads to a looser regret bound. This behavior is fully consistent with intuition: heavier concentration near the infimum naturally makes it harder to distinguish good regions, resulting in worse regret guarantees.
>
> (Q4) We appreciate this question on the distinction between theoretical regret and practical evaluation. To the best of our knowledge, almost all current bandit problems with delayed feedback measure regret on generated rewards (e.g., Joulani et al. (2013), Vernade et al. (2017), (Verma et al., 2022)), and hence we keep using this classic setting as the first exploration on Lipschitz bandits with delayed feedback. If the regret were measured only on the observed reward, the regret would be smaller (better) and hence still satisfies the claimed bound. Since the algorithms work by estimating the expected reward, they do not require modification to accommodate this change in the evaluation metric of regret.

---

> > ### Author Response · Authors · 2025-11-19
> >
> > (Q5) We sincerely thank the reviewer for this excellent question, which allows us to clarify a critical distinction between our model and the literature on bandits with censored feedback. While $P(\tau = \infty) > 0$ does introduce permanently missing observations, our setting is fundamentally different from the standard Censored Bandits model. In the censored feedback setting, the unobserved reward is typically censored to zero, which biases the empirical mean away from the true expected reward. In contrast, in our model, the reward $X_t$ (even if $\tau = \infty$) is simply unobserved (without assigning any value) but does not affect the calculation of the true underlying expected reward. Our analysis focuses on approximating this true $\mu(x)$ despite the unobserved samples, which distinguishes it from the regret bounds established for censored feedback.

---

### Official Review · Reviewer_QhYA · 2025-11-03

**Soundness:** 3
**Presentation:** 3
**Contribution:** 3
**Rating:** 6
**Confidence:** 3

**Summary:**

This paper studies the stochastic Lipschitz bandit problem with stochastic delayed feedback.
For the case of bounded delays, the authors extend the zooming algorithm to the delayed setting and obtain a regret bound of
$
\tilde{O}\left(T^{\frac{d_z+1}{d_z+2}} + \tau_{\max} T^{\frac{d_z}{d_z+2}}\right),
$
where $\tau_{\max}$ denotes the maximum delay.
For the unbounded-delay case, they propose a new algorithm that achieves nearly optimal regret bounds.
The paper further establishes instance-dependent lower bounds in the general unbounded-delay setting and presents empirical results that validate the theoretical findings.

**Strengths:**

1. The paper is well organised and  is easy to follow
2. The quantile-based upper bound and the matching lower bound (up to logs) are compelling and intuitive for unbounded delays.

**Weaknesses:**

1. Zooming algorithm is only proven for bounded delays, yet experiments show it works for unbounded delays. This gap is acknowledged but not resolved, leaving a significant theoretical question unanswered

**Questions:**

1. What are the main technical challenges in analysing the zooming algorithm under unbounded delays?
Empirically, the zooming algorithms seem to perform better than DLPP in both bounded and unbounded delays.

---

> ### Author Response · Authors · 2025-11-19
>
> We thank the reviewer for this highly insightful question. The core challenge in our theoretical analysis under unbounded delays revolves around bounding the number of missing observations. For any active arm $x$, the number of total pulls is $n(x) = v(x) + w(x)$, where $v(x)$ is the number of received observations and $w(x)$ is the number of missing observations. In the analysis of delayed Zooming algorithm, we bound $v(x)$ using Corollary 10 (line 652), and $w(x)$ is naturally upper bounded by $\tau_{\max}$ for bounded support. This allows us to derive the claimed regret bound, only requiring the bounded support assumption for generality. The main technical hurdle for unbounded delays is precisely bounding $w(x)$ without additional distributional assumptions about the delays. If we assume delays follow a specific distribution, such as sub-exponential delays, then $w(x)$ can be bounded using a high-probability tail bound. We believe analyzing this particular case is a promising direction and commit to exploring this in detail as future work.
>
> While the delayed Zooming Algorithm performs better for one dim rewards, it performs worse for two dim rewards. Due to the Pruning and Discretization strategy, DLPP performs better in higher dimensional cases.

---

### Official Review · Reviewer_R6E8 · 2025-11-04

**Soundness:** 3
**Presentation:** 3
**Contribution:** 3
**Rating:** 6
**Confidence:** 3

**Summary:**

The paper studies the standard Lipschitz bandit with stochastic delayed feedback. The paper gives theoretical results in bounded delay and unbounded delay scenarios. The paper conducts experiments to show the effectiveness of the proposed algorithms.

**Strengths:**

Originality: the exact problem setup has not been studied before to my knowledge.

Quality/clarity: I like the completeness of the paper. The setup is clear. The algorithm description is clear. The reason why the proof is difficult is clear and the experiments are quite comprehensive for a theoretical paper.

Significance: This paper paves a way for more studies in delayed feedback of Lipschitz bandit.

**Weaknesses:**

A minor thing: In related work there is a lot of relevant literature missing on bandits with delayed feedback. For example:

Thompson Sampling with Unrestricted Delays; Delay as Payoff in MAB; Impatient Bandits: Optimizing for the Long-Term Without Delay; Adaptivity and confounding in multi-armed bandit experiments

**Questions:**

1. For line 154 there are two $c$'s, one is $N_c(r)$ and one is the constant, maybe change this to make it less confusing?

---

> ### Author Response · Authors · 2025-11-19
>
> (Weakness) We thank the reviewer for highlighting these highly relevant omissions from the literature on bandits with delayed feedback. We have now thoroughly revised and expanded related work to include a discussion of more relevant work. Thanks a lot for helping improve the quality of our work.
>
> (Q1) Thank you for pointing it out. The $c$ in $N_c(r)$ is meant to stand for covering number, but it does confuse with the constant $c$. We have changed the notation to make it clear in the submitted revision.

---

### Meta-Review · Area_Chair_5qxh · 2026-01-06

**Summary:**

This paper studies the stochastic Lipschitz bandit problem with stochastic delayed feedback, both for the case where the delay is bounded and unbounded. The problem is relevant since both Lipschitz bandits and delayed feedback are natural in applications. The reviewers were overall positive and suggested unanimously acceptance of the paper.

**Reviewer Concerns:**

There were no major concerns and the rebuttal period only clarified some minor points.

**Reviewer Scores:**

I think that most reviewers would keep their score. Potentially, reviewer 2LpJ would go up from a 6 to a 7.

---

### Decision · Program_Chairs · 2026-01-26

Accept (Poster)